# Impact of Preprocessing Parameters in Medical Imaging-Based Radiomic Studies: A Systematic Review

**DOI:** 10.3390/cancers16152668

**Published:** 2024-07-26

**Authors:** Valeria Trojani, Maria Chiara Bassi, Laura Verzellesi, Marco Bertolini

**Affiliations:** 1Medical Physics, Azienda USL-IRCCS, 42123 Reggio Emilia, Italy; laura.verzellesi@ausl.re.it (L.V.); marco.bertolini@ausl.re.it (M.B.); 2Medical Library, Azienda USL-IRCCS, 42123 Reggio Emilia, Italy; mariachiara.bassi@ausl.re.it

**Keywords:** radiomics, preprocessing, biomarker, CT, MRI, CBCT, PET/CT

## Abstract

**Simple Summary:**

This review investigates how preprocessing parameters are related to the reproducibility and reliability of radiomic features derived from multimodality imaging techniques such as computed tomography (CT), magnetic resonance imaging (MRI), cone-beam CT (CBCT), and positron emission tomography (PET)/CT. Radiomics, which involves extracting quantitative features from medical images, shows great potential as a source of non-invasive clinical biomarkers but is hindered by variability in imaging parameters, especially during acquisition, reconstruction, and preprocessing. Standardizing and reporting preprocessing procedures is essential for consistent extraction of radiomic features, given their significant role in determining the robustness and reproducibility of these features.

**Abstract:**

Background: Lately, radiomic studies featuring the development of a signature to use in prediction models in diagnosis or prognosis outcomes have been increasingly published. While the results are shown to be promising, these studies still have many pitfalls and limitations. One of the main issues of these studies is that radiomic features depend on how the images are preprocessed before their computation. Since, in widely known and used software for radiomic features calculation, it is possible to set these preprocessing parameters before the calculation of the radiomic feature, there are ongoing studies assessing the stability and repeatability of radiomic features to find the most suitable preprocessing parameters for every used imaging modality. Materials and Methods: We performed a comprehensive literature search using four electronic databases: PubMed, Cochrane Library, Embase, and Scopus. Mesh terms and free text were modeled in search strategies for databases. The inclusion criteria were studies where preprocessing parameters’ influence on feature values and model predictions was addressed. Records lacking information on image acquisition parameters were excluded, and any eligible studies with full-text versions were included in the review process, while conference proceedings and monographs were disregarded. We used the QUADAS-2 (Quality Assessment of Diagnostic Accuracy Studies 2) tool to investigate the risk of bias. We synthesized our data in a table divided by the imaging modalities subgroups. Results: After applying the inclusion and exclusion criteria, we selected 43 works. This review examines the impact of preprocessing parameters on the reproducibility and reliability of radiomic features extracted from multimodality imaging (CT, MRI, CBCT, and PET/CT). Standardized preprocessing is crucial for consistent radiomic feature extraction. Key preprocessing steps include voxel resampling, normalization, and discretization, which influence feature robustness and reproducibility. In total, 44% of the included works studied the effects of an isotropic voxel resampling, and most studies opted to employ a discretization strategy. From 2021, several studies started selecting the best set of preprocessing parameters based on models’ best performance. As for comparison metrics, ICC was the most used in MRI studies in 58% of the screened works. Conclusions: From our work, we highlighted the need to harmonize the use of preprocessing parameters and their values, especially in light of future studies of prospective studies, which are still lacking in the current literature.

## 1. Introduction

In recent years, there has been a notable increase in research examining the potential of quantitative imaging features to reveal additional information from medical images beyond what is perceptible to the human eye [1,2,3]. Handcrafted radiomics systematically extracts quantitative imaging features from medical images to interpret biological information [4,5]. The term “radiomics” yields over 11,100 studies on PubMed. This approach takes manually delineated regions of interest, such as tumors, on medical images and extracts a high number of quantitative features with pre-determined mathematical formulas [6]. One of the biggest advantages of radiomics is that those values can be extracted from medical images which can be normally acquired from the clinical pathway of the patient. In radiomics studies, the hypothesis is that radiomic features (RFs) can act as clinical biomarkers, individually or in combination, even with clinical parameters [4].

Numerous investigations have explored and documented RFs’ ability to predict clinical outcomes, including both a prognostic aim (overall survival and treatment response) as well as diagnostic aim (e.g., differential diagnosis) [7,8,9,10,11,12,13,14,15,16,17,18].

As the field of radiomics has advanced, there has been increasing attention on its limitations [5,19]. The primary identified limitation is the sensitivity of RFs to variations in image acquisition and reconstruction parameters [20,21,22,23,24,25]. To be effective clinical biomarkers, RFs must demonstrate reproducibility across different imaging parameters for generalization [25]. However, numerous studies have documented differences in imaging acquisition and reconstruction parameters [25,26,27,28,29,30,31,32,33,34,35,36,37]. Moreover, the impact of variation in a single acquisition or reconstruction parameter on RF reproducibility depends on the individual feature [31,38]. Two works also highlight that reconstruction kernels notably affect RF reproducibility [21,37]. There are several works exploring various methods to address RF reproducibility across differently acquired scans, as well as how to standardize them.

Image preprocessing occurs between the image segmentation and feature extraction stages. There are several studies in the literature indicating that the repeatability of extracted radiomic features is significantly influenced by the image preprocessing settings employed [39,40,41,42,43,44]. Usually, the acquisition and reconstructed voxels are not isotropic, especially if no voxel re-segmentation is applied (e.g., CT scans are usually not isotropic at acquisition).

Normalization or range re-segmentation is used to remove voxels in the segmented volume of interest (VOI) that fall outside a specific gray-level range and is commonly used to ensure data consistency, especially for imaging modalities with calibrated units. In fact, normalization is typically necessary for CT and PET. However, this method does not apply to MRI data, which employ arbitrary intensity units instead. In this case, intensity outlier filtering is applied by calculating the mean (µ) and the standard deviation (σ) of gray levels within the VOI and excluding gray levels outside the range of µ ± n σ, where 3 is usually selected as the value of n [44,45,46,47]. Other relative normalizations have been proposed to determine MRI radiomic feature robustness [48,49]. While the results in the radiomic field are shown to be promising, these studies still have many pitfalls and limitations. One of the main issues in radiomic studies is that features depend on how the images are preprocessed before their computation. Preprocessing medical images with different parameters gives different feature values associated with the same image set, thus meaning that the associated results in terms of predictive model performances are also influenced. This work focused on gathering information about voxel resampling, normalization, and discretization preprocessing parameters from our included studies. We also investigated the study aim, anatomic district, and comparison metrics to better understand the up-to-date overview of radiomic studies. This review aims to provide an up-to-date and comprehensive overview of the most commonly used preprocessing techniques.

## 2. Materials and Methods

This systematic review followed the recommendations of the PRISMA-P (Preferred Reporting Items for Systematic Review and Meta-Analysis Protocols) 2020 [50]. The inclusion criteria for the records in the review process were the presence of specific preprocessing parameters evaluated in CBCT, CT, MRI, and PET/CT radiomic studies.

We performed a comprehensive literature search using four electronic databases: PubMed, Cochrane Library, Embase, and Scopus. Since radiomic studies are relatively recent, we conducted the literature search from January 2008 to December 2023. Mesh terms and free text were modeled in search strategies for databases using the one designed for PubMed: ((((“Positron-Emission Tomography” [Mesh]) OR “Tomography, X-ray Computed” [Mesh]) OR “Magnetic Resonance Imaging” [Mesh] OR PET OR Positron emission tomography OR computed tomography OR CT OR MRI OR magnetic resonance) AND (radiomics)) AND (preprocessing OR pre-processing). The inclusion criteria were studies with CT, PET/CT, or MRI as imaging modalities, where preprocessing parameters’ influence on feature values and model predictions was addressed. The study was exclusively confined to English-language research without publication status restrictions. The authors did not contact other institutions or researchers to uncover additional studies.

Two reviewers were involved in the study: a senior reviewer with 8 years of experience in the radiomic field of literature and a junior reviewer with 4 years of experience. These two reviewers independently screened records by reviewing their titles and abstracts and extracting general study characteristics (such as study title, first author’s name, publication year, journal, abstract, corresponding reviewer, and research keywords) for records meeting the inclusion criteria, using a customized data extraction form. In this first step, the authors independently used the ASReview tool (version 1.5) [51] to speed up the article selection. The final screening was manual; we used ASReview as a support tool. ASReview assigned a priority to the screened literature works, based on a subset of 4 relevant articles and 4 non-relevant articles (which were chosen accordingly by the two reviewers). After the tool scored the studies, we checked them in descending assigned priority order. Beyond a threshold priority, we confirmed that the works below that level were not relevant for the topic anymore and thus were excluded. The authors compared their decisions, resolving disagreements until a consensus was reached. Records lacking information on image acquisition parameters were excluded, and any eligible studies with full-text versions were included in the review process, while conference proceedings and monographs were disregarded. Additional articles not meeting the inclusion criteria upon full-text reading were identified and excluded at this stage. Reviewers independently selected full-text articles, resolving any uncertainties by reaching a consensus. Out of all the 53 full-text articles screened for inclusion, discussion to reach a consensus was required for 17 works (32%).

From each selected full-text article, the following parameters were identified and extracted:Acquisition modality (CBCT, CT, MRI, PET/CT);Number of patients or phantoms;Name of disease/s (if appropriate);Equipment vendor and model;Presence of acquisition parameters;Total number of features;Type of features subsampled in FO (first order), SM (shape metric), and TA textural features;Type of software used in the radiomic feature extraction;Image filtering used (Y/N; if Y, the type was reported);Voxel resampling;Normalization process;Discretization technique;Retrospective study (Y/N or NA);Statistical analysis: intraclass correlation coefficient (ICC), concordance correlation coefficient (CCC), area under the receiver operation curve (AUC), mean, average percentage difference, relative difference, Spearman correlation, Kolmogorov–Smirnov test, double-sample test, and two-way ANOVA;Type of study (reproducibility/repeatability/both or best performance);Main findings.

No contact with the authors of the records for complementary information was necessary. We synthesized our data in a table divided by the imaging modality subgroups.

Quality scoring for study selection was not utilized. The reviewers reviewed articles in duplicate, resolving disagreements by achieving an agreement. Data from the studies were standardized to mitigate potential biases. Data extracted from studies that did not meet the criteria for standardization were excluded.

This review aimed to describe the most used preprocessing pipeline in the radiomic workflow of the CBCT, CT, MRI, and PET/CT studies schematically shown in Figure 1. It also sought to find possible standard radiomic preprocessing setups and describe a potential evolution over time, aiming to uncover any increased awareness of the technique’s standardization.

Concerning the normalization strategies, we counted the number of papers that chose absolute, relative, or combination strategies. When this information was not applied, we set “None”. A normalization strategy enhances the robustness and reliability of the radiomic analysis. The absolute normalization strategy involves standardizing the intensity values of imaging by transforming them to a common scale based on absolute reference points. Conversely, relative normalization concerns adjusting the intensity values of images relative to a certain reference within each individual image or dataset; this technique often includes scaling the pixel values based on the mean (µ), median, or a specific percentile of the intensity values in the image. This approach accounts for differences in overall intensity levels between images by normalizing the intensities relative to their own distribution.

In terms of discretization strategies, we identified the number of studies that utilized a bin number (BN), a bin width (BW), or both the bin number and width (BN + BW) approach. When this information was not applied or not specified, we set “None”. The discretization strategy involves adjusting values to standardize the feature extraction process across different images or imaging modalities. The BN discretization ensures that the radiomic features are comparable and consistent by transforming the intensity values into a fixed number of discrete bins. The BW method assures that each bin represents a fixed range of intensity values. The BN + BW strategy is a dual approach that guarantees the simultaneous adjustment of the number of bins and the width of each bin used to discretize image intensity values.

These concepts were summarized in Figure 2.

We used the QUADAS-2 tool to investigate the risk of bias [52]. The tool evaluates four key domains: patient selection, index test, reference standard, and flow and timing. Each domain is evaluated for the risk of bias and applicability. We used this tool only for the first two domains because reference standards and flow and timing are not defined in this review’s field. Regarding the patient selection’s risk of bias, we checked the method used (lower marks were given in the case of inappropriate exclusions or not consecutive/random selection), while the applicability was evaluated if the included patients matched the typical clinical scenario.

We considered the index test the metric used in the result comparison. In this context, the risk of bias consisted of consistently evaluating the calculation/use of the aforementioned metric, and the applicability referred to the methodology of its use in practice.

## 3. Results

### 3.1. Literature Search

The computer-assisted search across PubMed, Cochrane Library, Embase, and Scopus yielded 546 records (as shown in Table 1). Following the elimination of duplicate entries, 459 unique records remained. Subsequently, a screening process based on the coherence of the title and abstract reduced the number of articles to 72 (refer to Figure 3).

Reviews and conference proceedings were excluded from further consideration. Consequently, 53 full-text articles underwent evaluation by the reviewers to determine their eligibility. Ten articles were excluded from the database for the following reasons: six lacked preprocessing results, one focused on a modality outside the scope of this review, two utilized only a digital phantom, and one was unfound. A comprehensive table with all the included entries can be found in the Appendix A.

### 3.2. Data Collection and Elaboration

The selected papers were grouped by considering the acquisition modality: one about CBCT, twelve about CT, twenty-six about MRI, and five about PET/CT.

CT studies were the most represented from 2013 to 2023. MRI works showed an increase in publications over the years, starting from one in 2017 to nine in 2023. PET/CT studies were published in 2019 (two) and 2021 (three).

In 2017, CT phantom studies were published to study the stability and reproducibility of radiomic features using phantoms made of different materials compared to those traditionally used in quality control (QC) [39,53,54].

The number of patients has increased, although not monotonically, from 256 in 2013 to 3000 in 2023. The modality that saw the greatest increase was MRI, demonstrating the growing interest of the scientific community in this type of imaging.

#### 3.2.1. Acquisition Parameter Presence and Voxel Resampling

Table 2 shows the studies reporting the acquisition parameters and the voxel resampling information. Only in PET and CBCT did we find that most papers used isotropic voxel resampling, while in CT and MRI, only 41.7% and 41.3% used it, respectively.For CT and CBCT, the most studied voxel resampling interpolation was 1 × 1 × 1 mm^3^. Furthermore, a significant number of studies used 1 × 1 × 1 mm^3^ in MRI, but more voxel sizes were investigated in the range of 0.9–4.8 mm^3^. This result is expected because in MRI the image characteristics are strongly influenced by the acquisition protocols.For PET/CT, voxel resampling dimensions were in the range of 1–4 mm^3^. This imaging modality, which employs a small resampling dimension, might introduce biases due to its intrinsic resolution (around non-isotropic 3–5 mm^3^).

#### 3.2.2. Normalization Strategies

Table 3 reports the normalization strategies adopted by the authors, considering the different modalities. In CT and PET/CT, most works did not use a normalization strategy, while in MRI, relative normalization was the most used, as expected. In CBCT, a relative strategy was used because the image gray levels are not in Hounsfield but relative units.

#### 3.2.3. Discretization Strategies

Most studies opted for a discretization strategy: 75% in CT, 100% in CBCT, 92.3% in MRI, and 100% in PET/CT. The combination of BN + BW was the most represented in this latter modality (Table 4).

In CT, the most investigated BNs were 32, 64, and 128, while BW ranged between 5 and 50 Hounsfield units (HU). In CBCT, the only work selected used BNs equal to 64, 128, and 256.

In MRI, the BN = 32 was the most used bin number, and eleven works studied it (42.3%), followed by BN = 64 and 128 (38.5% of the papers).

In PET/CT, the most studied BNs were 32 and 64, while BW typically ranged from 0.01 to 0.5 SUV.

#### 3.2.4. Study Aims

Table 5 shows the count of study aims for each imaging modality.

Most works employing CT and MRI aimed to study reproducibility and its association with repeatability. It is worth noting that since 2021, best performance has also been well investigated. Moreover, the CBCT work studied the aim of repeatability and reproducibility. PET/CT works consisted of three works about repeatability published in 2019 and 2021 and two works about best performance issued in 2021. Best performance means that the authors studied the influence of the preprocessing parameters based on the figure of merit of their chosen model, i.e., highlighting which preprocessing parameters yielded the best results in terms of performance of the model (AUC, accuracy, etc.); in fact, the highlighted studies in Section 3.2.5 use AUC as a comparison metric. The works under “best performance” aim to identify the set of preprocessing parameters that give the best performance with the predictive model they used.

#### 3.2.5. Anatomic District

Table 6 reports the anatomic districts categorized in the main four body regions. The CT studies focused on the trunk’s anatomical regions (chest, abdomen, and pelvis), while MRI studies were polarized between the brain and pelvis. In addition, CBCT and PET/CT investigations were centered on the pelvis or chest.

#### 3.2.6. Comparison Metrics

Table 7 illustrates the most common metrics used in the result comparison. In absolute terms, the most used metric was ICC, which is used predominantly in MRI, with 15 out of 26 works. Some works used more than one metric, so the number of papers reported in Table 7 is more than the number of works included.

### 3.3. Risk-of-Bias Analysis

The risk-of-bias assessment revealed that about 35% of the studies had a low risk of bias across the patient selection domain, while the remainder did not specify or were not clearly specified. In addition, 60% had a low risk of bias across the index test. The patient selection and index test applicability was higher than 72% (Figure 4).

## 4. Discussion

The volume of published studies on predictive modeling using radiomic features has been proliferating. However, no global consensus exists on which features are consistently repeatable and reproducible. This lack of agreement could potentially hinder future discussions on clinical applicability and the feasibility of prospective multi-institutional external validation trials.

Our work is a comprehensive image of the current literature, which highlights that there are indeed a range of preprocessing values that can be safely employed in radiomic studies without compromising the reliability of the obtained results.

We are aware that there are several papers published on US radiomics, since it is becoming a widely used imaging modality in medicine to investigate several diseases. Lately, the literature on US-based radiomics has been increasing [94,95,96]. In our study, we wanted to investigate 3D imaging, which is most used in oncologic and neurologic treatment, so we left US imaging out of our literature search.

This review’s main objective was to identify preprocessing strategies for radiomic features in the four primary 3D imaging modalities: CBCT, CT, MRI, and PET. It also aimed to identify common preprocessing parameter settings to evaluate the current state of the art in this field and raise awareness, thereby promoting more stable and reproducible results. In fact, the preprocessing technique has the greatest impact on feature reproducibility [58].

In 2017, CT phantom studies were published to study the stability and reproducibility of radiomic features using phantoms made of materials different from those traditionally used in QC [39,53,54]. Moreover, Palani et al. [63] published a phantom study in 2023, highlighting the scientific interest in this topic.

This review showed how the choice of 1 × 1 × 1 mm^3^ could be a feasible starting point for CT and MRI examinations if the original voxel size parameters are not too different from these values. On the other hand, for CT, the dimension related to slice thickness (the third dimension) is notably larger compared to the other two, which in turn depend only on the FOV and reconstruction matrix size (typically 512 × 512). Also, for this reason, Larue et al. [53] and Shafiq-ul Hassan [39] studied 1 × 1 × 3 and 1 × 1 × 2 mm^3^, respectively. In MRI, however, the discussion is broader. From a radiomic analysis perspective, a volumetric acquisition might be the best choice to introduce minimal bias in voxel resampling. However, this choice is not necessarily optimal for all anatomical regions/lesions, and in any case, it does not apply to retrospective studies. For PET/CT, it seems clear that an isotropic voxel of approximately 3 × 3 × 3 or 4 × 4 × 4 mm^3^ is a widely accepted choice and a good compromise for obtaining comparable results. The challenges in isotropic reconstruction are similar to those previously described for CT, even though they relate to different physical processes.

A limitation of this review is that it did not delve into the issue of the resampling function used (e.g., linear, bicubic, nearest-neighbor interpolation, etc.) because the studies were too varied to obtain a meaningful result. Larue et al. [53] demonstrated that linear interpolation resulted in the narrowest range of feature values for nearly half of the features, and cubic interpolation did so for a smaller portion. In contrast, nearest-neighbor interpolation produced the broadest range for most CT study features.

Other limitations of this study lay in the impossibility extracting from the screened literature works a definite set of recommended values to use for the preprocessing step of radiomic works. Furthermore, the inclusion of a third reviewer to resolve the disputes and the lack of consensus would have been beneficial for our work.

Different techniques have been considered for normalization strategies. Starting from CT, Larue et al. [53] studied the influence of BW from 5 to 50 HU at 5 HU intervals. Fave et al. [61] preprocessed the CT images with 8-bit depth resampling using a 16 HU bin width. Kolossváry et al. [60] compared two equally sized bins and equally probable bins where each bin contains a proportion of the data, finding that all GLRLM features were significantly affected by binning type; in addition, BNs significantly affected the values for all GLCMs and GLRLMs. Gray-level resampling affects only second- and higher-order radiomics features, while voxel size variation can influence first-, second-, and higher-order features [39]. The optimal discretization methods for feature inter- and intra-sample reproducibility depend on the imaging modality. The choice of discretization significantly affects intensity distributions, feature values, and reproducibility [45]. Defining only bin number or width is insufficient, as these two quantities are related to the maximum number of gray levels used. The IBSI group mentions preprocessing recommendations in their work [45]. In modalities that use calibrated imaging intensity units (e.g., CT and PET/CT), when a re-segmentation range is defined, either fixed bin size or fixed bin width can be defined. If no re-segmentation range is used in calibrated imaging modalities, then fixed bin number is recommended. When a non-calibrated imaging modality (e.g., MRI) is employed, they recommend using the fixed bin number method without intensity re-segmentation. The issue when using BN is that it introduces a normalization effect, directly comparing feature values across patients, whereas BW preserves the relationship between PET units and their physical meaning. Xu et al. [91] indicate that the BN discretization scheme achieves relatively high stability compared to BW, in line with IBSI recommendations [45], contrasting with Pfaehler et al. [28].

Most CT and MRI studies have focused on reproducibility and its relationship with repeatability. Notably, since 2021, there has been significant investigation into achieving optimal performance. Additionally, CBCT studies have examined both repeatability and reproducibility. PET/CT research includes three studies on repeatability published in 2019 and 2021, and two studies on optimal performance published in 2021. The selection of the best performance as the criterion for evaluating the quality of a model (including the choice of preprocessing parameters) should be limited, as this choice could introduce biases.

About the anatomical part investigated, CT studies focused on the torso’s anatomical regions (chest, abdomen, and pelvis), while MRI studies examined the brain and pelvis regions. Additionally, CBCT and PET/CT investigations were centered on the pelvis or chest. The distribution of the anatomical regions per imaging modality aligns with what is expected in a therapeutic pathway (e.g., using MRI for further investigation of the brain or prostate/cervix due to its capability to distinguish between soft tissues compared to CT).

Concerning comparison metrics, the most commonly used metric was ICC, predominantly applied in MRI studies, with 15 out of 26 works [65,66,67,68,71,73,74,79,82,83,84,85,86,88,89]. ICC is extensively studied in radiomics research due to its role in quantitative assessment, reliability, wide applicability, and standard practice. It serves as a quantitative measure to gauge agreement or consistency between measurements or observations, crucial for evaluating the reproducibility of radiomic features. ICC evaluates measurement reliability by accounting for systematic and random variations, making it effective for assessing feature stability across different scans or observers. Its versatility extends to various data types, including continuous and categorical variables, enabling the evaluation of different radiomic features across diverse imaging modalities and patient groups. Overall, ICC is favored in radiomics for its comprehensive and standardized approach to assessing feature reproducibility and stability, thereby supporting the development and validation of radiomics as a quantitative imaging biomarker. CCC is another essential statistical metric used in radiomics research to evaluate measurement agreement and reliability. It quantifies the agreement between two sets of continuous measurements, assessing precision (how closely measurements cluster around the line of perfect agreement) and accuracy (how close measurements are to the line of identity), offering a comprehensive measure of agreement that is easy to interpret. Like ICC, CCC accounts for systematic bias (shift) and random variation (spread) between measurements. This makes it suitable for evaluating the reproducibility and stability of radiomic features across different scans or observers. AUC quantifies the overall discriminatory ability of a diagnostic test or biomarker. In radiomics, this metric assesses the ability of radiomic features or models to distinguish between different clinical conditions or outcomes based on imaging data. Higher AUC values indicate better discriminatory performance of the radiomic feature or model.

From our investigated studies, it was not possible to identify a recommended set of preprocessing parameters or a subset of stable features due to the variability in the experimental setups. However, due to their definitions, higher-order class features (e.g., GLRM and GLCM-based) are more affected by preprocessing parameters as well as acquisition and reconstruction parameters, so they should be used with caution when dealing with retrospective multicentric studies. We can conclude that, when building a predictive model, first-order class features should be preferred for inclusion should they show a correlation with the chosen outcome, especially if a preprocessing strategy is not employed.

Regarding the risk of bias analysis, it was not possible to assess it for the majority of the studies for patient selection. Since most of the radiomic studies are retrospective, it is not possible to understand how the patient selection process might affect the results. Many works showed low concern for risk of bias in the index test, due to the widely accepted choice of well-known statistical indexes, already broadly used in the literature to test repeatability even in other scientific fields. Regarding applicability, most studies raised low concern for both patient selection and the index test.

From our analysis, we also found out that a significant number of our included studies pursued best model performance in order to find the optimal preprocessing parameters [55,56,59,70,72,75,76,77,80,82,90,91]. While this is still a feasible approach given the early stage of the literature and research in this field, it would be advised to strive toward a standardization that is based on the feature stability rather than individual model performances as these are influenced by other variables (such as training/validation datasets).

The problem of generalizability remains open, but using a preprocessing strategy improves the robustness of the analysis, especially when multiple vendors and machine models are included.

We think that an effort toward the standardization of preprocessing parameters should be conducted by the scientific community. For example, the IBSI group has taken a step in that direction. From this work, we can see that there are several sets of preprocessing values that are employable; with such initiatives, a consensus on this topic can be reached.

## 5. Conclusions

To date, it does not seem possible to establish a standardized recipe for the unequivocal selection of preprocessing parameter combinations for radiomic analysis. However, it is noteworthy that certain combinations of voxel resampling, normalization, and digitalization are more commonly used than others.

For reproducibility, a study must report the sequence of preprocessing parameters used so that the methodology employed is evident and transparent.

Our work is a comprehensive image of the current literature which highlights that there are indeed a range of preprocessing values that can be safely employed in radiomic studies without compromising the reliability of the obtained results.

From our work, we highlighted the need to harmonize the use of preprocessing parameters and their values, especially considering future studies of prospective studies, which are still lacking in the current literature.

## Figures and Tables

**Figure 1 cancers-16-02668-f001:**
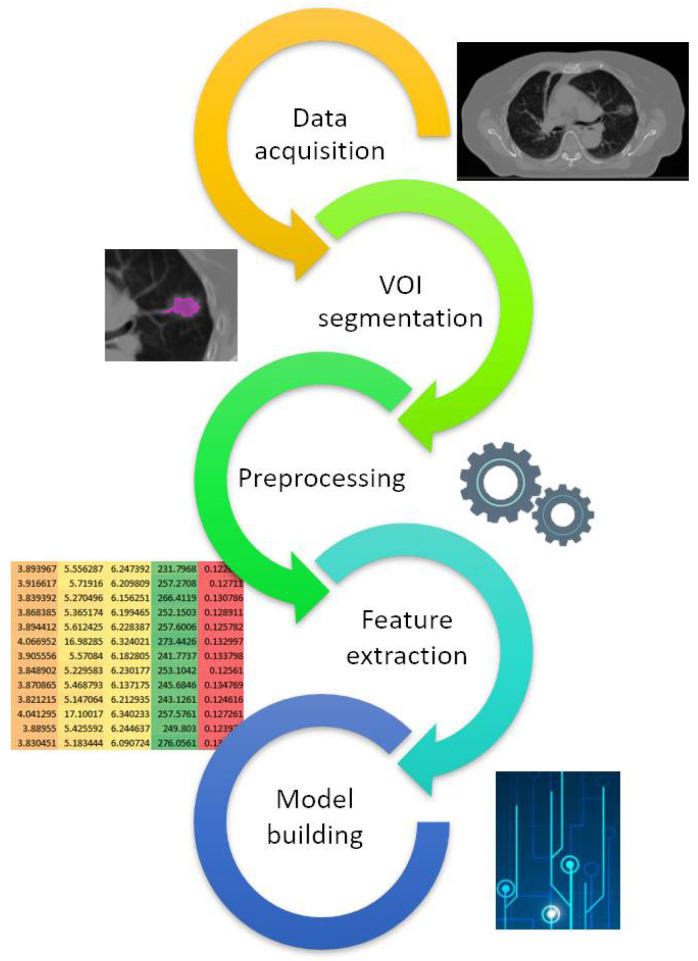
Schematic radiomic workflow. Regarding voxel resampling, we counted the number of works that used single isotropic, multiple isotropic, and non-isotropic voxel interpolation. When the original voxel size was used, we put “N.A.”, and when the information was not indicated, “None”.

**Figure 2 cancers-16-02668-f002:**
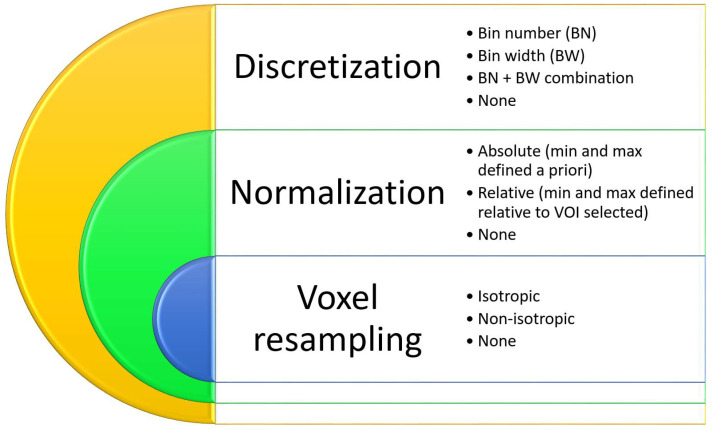
Preprocessing scheme used in the evaluation of the items. In addition, we investigated the study aims, the anatomical districts considered, and the metric used within the works. Concerning the study aims, we counted how many studies used best performance, repeatability, or both strategies. The anatomical districts were categorized into “abdomen”, “brain”, “thorax”, and “pelvis” for simplicity’s sake. When no anatomical part was specified, e.g., for phantom studies, we set “N.A.”. Moreover, the metric indexes were grouped into ICC, CCC, AUC, or “Other” when different statistical metrics were used.

**Figure 3 cancers-16-02668-f003:**
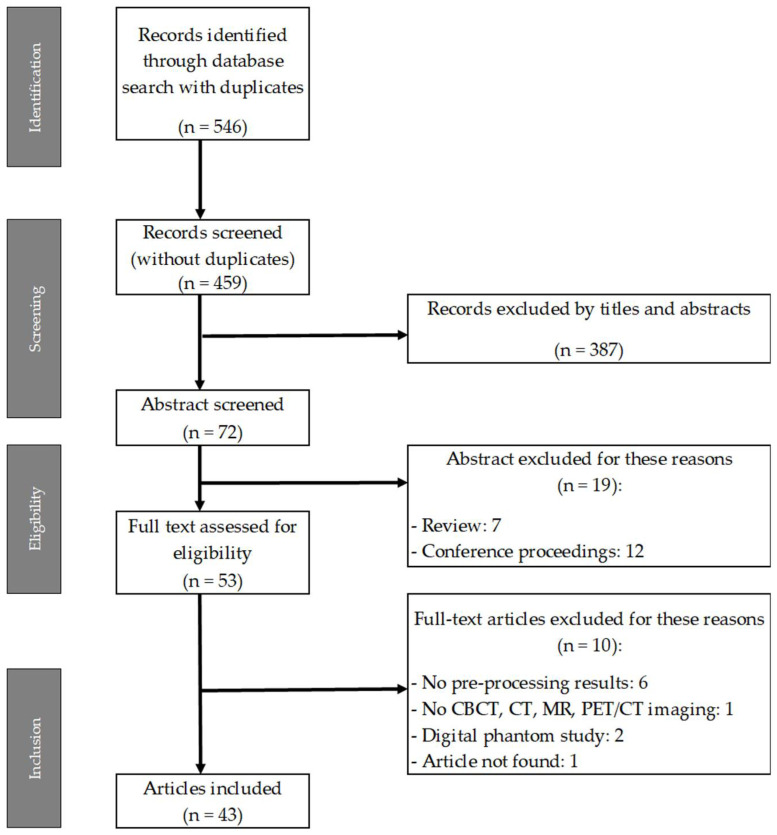
PRISMA-P flow chart: article selection process.

**Figure 4 cancers-16-02668-f004:**
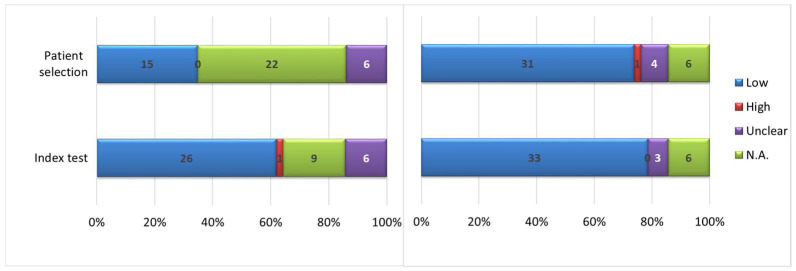
Risk of bias (**left**) and applicability concerns (**right**).

**Table 1 cancers-16-02668-t001:** The distribution of found records across databases.

Database (n° of Record)				n° of Total Records (with Duplicates)	n° of Total Records (without Duplicates)
Medline	Embase	Cochrane	Scopus		
208	286	11	41	546	459

**Table 2 cancers-16-02668-t002:** The number of articles included and stratified by voxel resampling type (N.A. indicates this information is missing).

Modality	Acquisition Parameter Reporting	Voxel Resampling				
		Isotropic	Multiple Isotropic	Non-Isotropic	N.A.	None
CT	10 (83.3%)	2 (16.7%)	3 (25.0%)	2 (16.7%)	5 (41.7%)	0 (0.0%)
Ref.		[55,56]	[54,57,58]	[39,53]	[59,60,61,62,63]	
CBCT	1 (100%)	1 (100%)	0 (0%)	0 (0%)	0 (0%)	0 (0%)
Ref.		[64]				
MRI	21 (80.8%)	9 (34.6%)	2 (7.7%)	3 (11.5%)	5 (19.2%)	7 (26.9%)
Ref.		[56,65,66,67,68,69,70,71,72]	[73,74]	[75,76,77]	[78,79,80,81,82]	[83,84,85,86,87,88,89]
PET/CT	5 (100%)	2 (40%)	2 (40%)	0 (0%)	0 (0%)	1 (20%)
Ref.		[90,91]	[28,92]			[93]

**Table 3 cancers-16-02668-t003:** Normalization strategies grouped by modality. Combination stands for absolute and relative strategies used for comparison.

Modality	Absolute	Relative	Combination	None
CT	0 (0%)	1 (8.3%)	1 (8.3%)	10 (83.3%)
Ref.		[56]	[58]	[39,53,54,55,57,59,60,61,62,63]
CBCT	0 (0%)	1 (100%)	0 (0%)	0 (0%)
Ref.		[64]		
MRI	0 (0%)	14 (53.8%)	4 (15.4%)	8 (30.8%)
Ref.		[56,68,71,74,76,77,79,80,81,82,83,84,87,88]	[65,66,67,85]	[69,70,72,73,75,78,86,89]
PET/CT	0 (0%)	1 (20%)	1 (20%)	3 (60%)
Ref.		[90]	[92]	[28,91,93]

**Table 4 cancers-16-02668-t004:** Discretization strategies grouped by modality.

	BN	BW	BN + BW	None
CT	2 (16.7%)	6 (50%)	1 (8.3%)	3 (25%)
Ref.	[39,59]	[53,55,56,57,58,61]	[60]	[54,62,63]
CBCT	1 (100%)	0 (0%)	0 (0%)	0 (0%)
Ref.	[64]			
MRI	10 (38.5%)	9 (36.6%)	5 (19.2%)	2 (7.7%)
Ref.	[65,69,71,75,81,82,84,85,86,87]	[56,66,72,73,76,77,83,88,89]	[67,68,74,78,79]	[70,80]
PET/CT	1 (20%)	0 (0%)	4 (80%)	0 (0%)
Ref.	[92]		[28,90,91,93]	

**Table 5 cancers-16-02668-t005:** The study aims per modality.

	Best Performance	Repeatability	Reproducibility	Repeatability + Reproducibility
CT	3 (25%)	2 (16.7%)	5 (41.6%)	2 (16.7%)
Ref.	[55,56,59]	[54,62]	[39,57,58,60,61]	[53,63]
CBCT	0 (0%)	0 (0%)	0 (0%)	1 (100%)
Ref.				[64]
MRI	8 (30.8%)	7 (26.9%)	6 (23.1%)	5 (19.2%)
Ref.	[56,70,72,75,77,78,80,82]	[66,73,76,84,86,87,88]	[65,67,68,74,79,81]	[69,71,83,85,89]
PET/CT	2 (40%)	3 (60%)	0 (0%)	0 (0%)
Ref.	[90,91]	[28,92,93]		

**Table 6 cancers-16-02668-t006:** Investigated “macro” anatomic districts divided per modality.

	Abdomen	Brain	Thorax	Pelvis	N.A.
CT	1 (8.4%)	0	3 (25%)	4 (33.3%)	4 (33.3%)
Ref.	[55]		[54,61,62]	[57,58,59,60]	[39,53,56,63]
CBCT	0	0	0.0%	1 (100%)	
Ref.				[64]	
MRI	2 (7.7%)	10 (38.5%)	2 (7.7%)	9 (34.6%)	4 (15.3%)
Ref.	[76,78]	[65,67,69,74,75,80,81,82,84,86]	[72,73]	[66,68,70,75,77,79,85,88,89]	[56,71,83,87]
PET/CT	0	0	3 (60%)	2 (40%)	0
Ref.			[28,91,93]	[90,92]	

**Table 7 cancers-16-02668-t007:** Comparison metrics per imaging modality.

	ICC	CCC	AUC	Other
CT	3	3	2	4
Ref.	[57,58,60]	[53,54,62]	[56,59]	[39,55,61,63]
CBCT	0	1	0	0
Ref.		[64]		
MRI	15	7	5	9
Ref.	[65,66,67,68,71,73,74,79,82,83,84,85,86,88,89]	[67,76,83,85,86,87,89]	[56,70,77,80,82]	[68,69,71,72,75,78,81,84,85]
PET/CT	2	1	1	1
Ref.	[28,93]	[92]	[90]	[91]

## Data Availability

The data were obtained from a computer-assisted search of PubMed, Cochrane Library, Embase, and Scopus.

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
