# Peer review of "Impact of Preprocessing Parameters in Medical Imaging-Based Radiomic Studies: A Systematic Review"

_cancers, 2024, doi:10.3390/cancers16152668_

Round 1

Reviewer 1 Report

Comments and Suggestions for Authors

I have the following comments:

1) For greater clarity and exhaustiveness, the Abstract should be written in a structured format (e.g., Background/Objectives, Methods, Results and Conclusions) and summarize the main quantitative findings of the systematic review performed by the authors.

2) Introduction. It is too long and should be shortened by about 25%, e.g., by reducing and partially merging together the paragraphs between lines 54 and 64 and those between lines 65 and 86. The paragraph between lines 47 and 53 contains information that is partially redundant with the preceding paragraph and quite obvious for most readers, and hence could be eliminated altogether.

3) Materials and methods. At line 108, please report the degree and type of experience of the two reviewers (e.g., in terms of years after board certification and/or of subspecialty professional activity). Moreover (line 119), please specify in which proportion of cases consensus reading was required.

4) Results. Generally OK.

5) Discussion. From line 353 to the end of the Discussion section (line 420), no references to the literature searched appear. Please add relevant citations along with specific statements that summarize the authors' comments as supported by the literature, so as to avoid generic comments without literature support.

6) Overall, the English language of the manuscript should undergo moderate editing, ideally with the aid of a native English-speaking medical writer or a professional language editing service.

Comments on the Quality of English Language

Moderate English language revision required.

Author Response

I have the following comments: 

  1. For greater clarity and exhaustiveness, the Abstract should be written in a structured format (e.g., Background/Objectives, Methods, Results and Conclusions) and summarize the main quantitative findings of the systematic review performed by the authors. 

We thank the referee for this comment. We changed the abstract accordingly. 

  1. Introduction. It is too long and should be shortened by about 25%, e.g., by reducing and partially merging together the paragraphs between lines 54 and 64 and those between lines 65 and 86. The paragraph between lines 47 and 53 contains information that is partially redundant with the preceding paragraph and quite obvious for most readers, and hence could be eliminated altogether. 

We thank the referee for this comment. We tried reducing the introduction section as suggested, removing the redundant descriptions. 

  1. Materials and methods. At line 108, please report the degree and type of experience of the two reviewers (e.g., in terms of years after board certification and/or of subspecialty professional activity). Moreover (line 119), please specify in which proportion of cases consensus reading was required. 

The senior reviewer had 8 years of experience and the junior reviewer had 4 years of experience in the radiomic field.  Out of all the 53 full-text articles screened for inclusion, the two reviewers had to discuss and reach a consensus for 17 works (32%). We added this information to the manuscript. 

  1. Results. Generally OK. 

Thank you. 

  1. Discussion. From line 353 to the end of the Discussion section (line 420), no references to the literature searched appear. Please add relevant citations along with specific statements that summarize the authors’ comments as supported by the literature, so as to avoid generic comments without literature support. 

We included the reference to the searched literature for the statements supporting them. The other statements are the authors’ inference which was then pointed out in the Discussion section accordingly. 

  1. Overall, the English language of the manuscript should undergo moderate editing, ideally with the aid of a native English-speaking medical writer or a professional language editing service. 

We revised the manuscript. 

Reviewer 2 Report

Comments and Suggestions for Authors

The authors investigate how preprocessing parameters are related to the reproducibility and reliability of radiomic features derived from multimodality imaging techniques such as CT, MRI, CBCT, and PET/CT. 

The aim is really interesting because the radiomic represents a possible evolution of radiological science. 

The article is well written and I have two suggestion:

- describe better the possible use of radiomics in clinical radiological practice

- add a "statistical analysis" section

- add some figures-graphs

- table 1 is necessary?

- table 5 line two lack somes "0"

Author Response

The authors investigate how preprocessing parameters are related to the reproducibility and reliability of radiomic features derived from multimodality imaging techniques such as CT, MRI, CBCT, and PET/CT.  

The aim is really interesting because the radiomic represents a possible evolution of radiological science.  

The article is well written and I have two suggestion: 

- describe better the possible use of radiomics in clinical radiological practice 

We thank the referee for this comment. Does the referee mean how preprocessing might affect clinical radiological practice? One limitation of our study was the lack of a fixed set of recommended preprocessing values, but we highlighted that specific parameter values are used in the majority of works for the different modalities.  

Our work is a comprehensive image of the current literature, which highlights that a range of preprocessing values can be safely employed in radiomic studies without compromising the reliability of the obtained results. 

We added a sentence to the conclusion section to highlight this issue. 

- add a "statistical analysis" section 

 Since we did not perform a statistical analysis but only a review of the current literature, we did not add the dedicated section. 

- add some figures-graphs 

 Following the Reviewer’s suggestion, we added two additional figures: Fig.1 to describe the radiomic process in a visual language schematically and Fig. 2 to explain better, also from a graphic point of view, the summarization of the preprocessing evaluated within this systematic review. 

- table 1 is necessary? 

We thought it was a concise way to summarize the distribution of the found articles in the different databases due to the nature of the systematic review and the requests from PRISMA statement for systematic reviews. 

- table 5 line two lack somes "0" 

We thank the referee for this comment. We fixed it. 

Reviewer 3 Report

Comments and Suggestions for Authors

Thanks for sharing this useful study. Here are my comments.

1. Why ultrasound radiomics is not included in this study?

2. Major findings should be highlighted in the beginning of the discussion section.

3.. How preprocessing affect the performance with respect to each modalities in term of various measurement metric such as AUC should be provided in introduction section.

4. This study lacks of good figures and image samples.

5. What does it mean by the “best performance” in line 276? It should be well-defined.

Author Response

Thanks for sharing this useful study. Here are my comments. 

  1. Why ultrasound radiomics is not included in this study?

We are aware that there are several papers published on US radiomics, since it is becoming a widely used imaging modality in medicine to investigate several diseases. Lately, the literature on US–based radiomics is increasing [https://doi.org/10.1016/j.diii.2021.10.004, https://link.springer.com/article/10.1007/s00330-022-08662-1. (https://doi.org/10.1007/s00330-022-08662-1, http://dx.doi.org/10.11152/mu-3248 ]. In our study, we wanted to investigate 3D imaging, which is most used in oncologic and neurologic treatment, so we left US imaging out of our literature search.  

We added this sentence to the Discussion section, line 355, to further clarify this point. 

  1. Major findings should be highlighted in the beginning of the discussion section.

 We thank the Reviewer for this observation. We added the following sentence, “Our work is a comprehensive image of the current literature which highlights that there are indeed a range of preprocessing values that can be safely employed in radiomic studies without compromising the reliability of the obtained results.” 

We added a sentence to the Discussion and Conclusion section to highlight this issue. 

3.. How preprocessing affect the performance with respect to each modalities in term of various measurement metric such as AUC should be provided in introduction section. 

The analysis is carried out in the results section because it was the aim of this work. Still, we added this sentence, “Preprocessing medical images with different parameters give different feature values associated to the same image set, thus meaning that the associated results in terms of predictive model performances are also influenced.” to the introduction section, line 106, to clarify this point further.  

  1. This study lacks of good figures and image samples.

 We thank the Reviewer for the comment; we added two additional figures: Fig.1 to describe the radiomic process in a visual language schematically and Fig. 2 to explain better, also from a graphic point of view, the summarization of the preprocessing evaluated within this systematic review.  

  1. What does it mean by the “best performance” in line 276? It should be well-defined.

Best performance means that the authors studied the influence of the preprocessing parameters based on the figure of merit of their model of choosing, i.e. highlighting which preprocessing parameters yielded the best results in terms of performance of the model (AUC, accuracy, ...), in fact the highlighted studies in section 3.2.5 use AUC as a comparison metric. The works under the “best performance” aim to identify the set of preprocessing parameters that give the best performance with the predictive model they used. 

We added this sentence to the respective section to clarify this point better. 

Reviewer 4 Report

Comments and Suggestions for Authors

The manuscript is a systematic review about the impact of preprocessing parameters on the reproducibility and reliability of radiomic features in multimodality imaging (CT, MRI, CBCT, PET/CT).

My comments:

Simple Summary and Abstract:

Spell out all acronyms the first time that they are used (e.g.: CT, MRI,CBCT)

Materials and Methods:

 Why were specialized databases, such as Web of Science, were not included in the literature search?

Would you consider including a validation step for the automated screening tool (ASReview) to ensure its decisions align with manual screening?

Discussion:

The limitations of the study are not clearly defined in the Discussion section.

Conclusions:

The conclusion acknowledges the challenge of establishing standardized preprocessing parameters for radiomic analysis, but it does not provide clear instructions on how to address this issue.   

Author Response

The manuscript is a systematic review about the impact of preprocessing parameters on the reproducibility and reliability of radiomic features in multimodality imaging (CT, MRI, CBCT, PET/CT). 

My comments: 

Simple Summary and Abstract: 

Spell out all acronyms the first time that they are used (e.g.: CT, MRI,CBCT) 

We modified the simple summary accordingly.  

 Materials and Methods: 

 Why were specialized databases, such as Web of Science, were not included in the literature search? 

We performed the literature search on 4 databases: PubMed, Cochrane Library, Embase, and Scopus, while usually for literature searches 3 is the minimum, we didn’t think to include even another database. Furthermore, Web of Science, as well as Scopus, is not a specialized database like PubMed.  

Would you consider including a validation step for the automated screening tool (ASReview) to ensure its decisions align with manual screening? 

The screening was manual, we used ASReview as a support tool. ASReview assigned a priority to the screened literature works, based on a subset of 4 relevant articles and 4 non-relevant articles (which were chosen accordingly by the two reviewers). After the tool scored the studies, we checked them in descending assigned priority order. Beyond a threshold priority, we confirmed that the works below that level were not relevant for the topic anymore and thus were excluded.  

We changed the manuscript (materials & methods) to clarify this aspect. 

Discussion: 

The limitations of the study are not clearly defined in the Discussion section. 

Besides the limitations already included in the discussion (line 367), one of the limitations of this study lied in the impossibility to extract from the screened literature works a definite set of recommended values to use for the preprocessing step of radiomic works. Furthermore, the inclusion of a third reviewer to resolve the disputes and the lack of consensus would have been beneficial for our work. 

We added this sentence to the manuscript (discussion, line 373) to clarify this point. 

Conclusions: 

The conclusion acknowledges the challenge of establishing standardized preprocessing parameters for radiomic analysis, but it does not provide clear instructions on how to address this issue.   

We thank the referee for this comment. The lack of a fixed set of preprocessing recommended values was one of the limitations of our study but we highlighted that there are specific parameters values that are used in the majority of works for the different modalities.  

Our work is a comprehensive image of the current literature which highlights that there are indeed a range of preprocessing values that can be safely employed in radiomic studies without compromising the reliability of the obtained results. 

We added a sentence to the conclusion section to highlight this issue.  

Reviewer 5 Report

Comments and Suggestions for Authors

Manuscript ID: cancers-3093797
Type of manuscript: Review
Title: Impact of preprocessing parameters in multimodality imaging-based radiomic studies: a review

This manuscript provides a systematic review on how image preprocessing parameters impact the reproducibility and reliability of radiomics features derived from medical imaging modalities including CT, MRI, CBCT, and PET/CT. This is interesting and meaningful. While the manuscript is well written, the following concerns need to be addressed before publication.

1.    In writing this review, why the authors did not consider ultrasound? There should be many papers published related to ultrasound radiomics.
2.    The authors used ‘multimodality imaging’ in the title and the text. This is quite confusing, because multimodality imaging generally refers to employing two or more imaging modalities in a single study. However, what the authors really mean is different medical imaging modalities, which can be confirmed in the search strategy, where there is no ‘multimodality.’ Therefore, I would suggest the authors change the ‘multimodality imaging’ to ‘medical imaging.’
3.    Line 115. Search strategy: ‘preprocessing OR preprocessing.’ Why there are two repeated ‘preprocessing’? Please check this.
4.    The limitations of this systematic review should be clarified in Discussion.
5.    It would be better to discuss the future developments in Discussion.
6.    A long table comparing the 43 papers included in this systematic review is suggested. The comparison should include several aspects, meaning that the table should have several columns.
7.    Title. This is a systematic review. The title is suggested to be changed from ‘a review’ to ‘a systematic review.’

Author Response

Type of manuscript: Review 

Title: Impact of preprocessing parameters in multimodality imaging-based radiomic studies: a review 

We think that this reviewer has not read the latest version of the manuscript we submitted. The new submitted manuscript is the latest version with their corrections as well. 

This manuscript provides a systematic review on how image preprocessing parameters impact the reproducibility and reliability of radiomics features derived from medical imaging modalities including CT, MRI, CBCT, and PET/CT. This is interesting and meaningful. While the manuscript is well written, the following concerns need to be addressed before publication. 

  1. In writing this review, why the authors did not consider ultrasound? There should be many papers published related to ultrasound radiomics.

We are aware that there are several papers published on US radiomics, since it is becoming a widely used imaging modality in medicine to investigate several diseases. Lately, the literature on US –based radiomics is increasing [https://doi.org/10.1016/j.diii.2021.10.004, https://link.springer.com/article/10.1007/s00330-022-08662-1. (https://doi.org/10.1007/s00330-022-08662-1, http://dx.doi.org/10.11152/mu-3248 ]. In our study, we wanted to investigate 3D imaging, which is most used in oncologic and neurologic treatment, so we left US imaging out of our literature search.  

We added this sentence to the Discussion section, line 355, to further clarify this point. 

  1. The authors used ‘multimodality imaging’ in the title and the text. This is quite confusing, because multimodality imaging generally refers to employing two or more imaging modalities in a single study. However, what the authors really mean is different medical imaging modalities, which can be confirmed in the search strategy, where there is no ‘multimodality.’ Therefore, I would suggest the authors change the ‘multimodality imaging’ to ‘medical imaging.’

We thank the referee for this comment. Even though some of the included studies had  two or more imaging modalities, we still screened also studies with a single modality. We changed the title accordingly.  

  1. Line 115. Search strategy: ‘preprocessing OR preprocessing.’ Why there are two repeated ‘preprocessing’? Please check this.

This was a typo, we meant pre-processing. We fixed it. 

  1. The limitations of this systematic review should be clarified in Discussion.

Besides the limitations already included in the discussion (line 358), one of the limitations of this study lied in the impossibility to extract from the screened literature works a definite set of recommended values to use for the preprocessing step of radiomic works. Furthermore, the inclusion of a third reviewer to resolve the disputes and the lack of consensus would have been beneficial for our work. 

We added this sentence to the manuscript (discussion, line 364) to clarify this point. 

  1. It would be better to discuss the future developments in Discussion.

We think that an effort toward the standardization of preprocessing parameters should be conducted by the scientific community. For example, the IBSI group has taken a step in that direction. From this work, we can see that there are several sets of preprocessing values that are employable, with such initiatives a consensus on this topic can be reached. 

We added this sentence on the Discussion section of the manuscript. 

  1. A long table comparing the 43 papers included in this systematic review is suggested. The comparison should include several aspects, meaning that the table should have several columns.

We initially made this table, but we thought it wouldn’t fit weel in the manuscript. We can add it as a supplementary material for completeness. 

  1. Title. This is a systematic review. The title is suggested to be changed from ‘a review’ to ‘a systematic review.’

We have already changed the title in our latest version. We thank the referee for this comment. 

Round 2

Reviewer 1 Report

Comments and Suggestions for Authors

Thank you. No further comments.

Comments on the Quality of English Language

Some additional, minor English language editing should be performed.

Reviewer 4 Report

Comments and Suggestions for Authors

The authors have satisfactorily addressed my concerns.

Reviewer 5 Report

Comments and Suggestions for Authors

The revision has addressed my concerns. Thanks.